# Linear Resonator Actuator-Constructed Wearable Haptic System with the Application of Converting Remote Grinding Force to Vibratory Sensation

**Shang-Hsien Liu** [1], **Yung-Chou Kao** [2] **and Guo-Hua Feng** [1,3,*]

1 Department of Power Mechanical Engineering, National Tsing Hua University, Hsinchu 300044, Taiwan; s110033536@m110.nthu.edu.tw
2 Advanced Institute of Manufacturing with High-Tech Innovations, and Department of Mechanical Engineering, National Chung Cheng University, Chiayi 621301, Taiwan; imeyckao@ccu.edu.tw
3 Institute of Nano Engineering and MicroSystems, National Tsing Hua University, Hsinchu 30013, Taiwan
* Correspondence: ghfeng@pme.nthu.edu.tw

**Abstract:** This study developed a three-axis vibrational haptic wearable device (RCWS) utilizing Linear Resonant Actuators (LRAs) to simulate grinding vibrations. The implementation of RCWS is described in detail. By recording the normal force during manual grinding with a load cell and converting it into a series of PWM commands, the LRA on the RCWS was controlled in open-loop mode using these PWM commands. Three methods were tested for force-to-PWM conversion, two of which showed a linear correlation (>0.7) with raw data. In the correlation between PWM commands and generated acceleration, all three methods exhibited a high linearity of at least 0.85. This wearable RCWS offers a promising approach for users to experience the machining force from the versatile and critical remote machining process with a finger vibratory sensation.

**Keywords:** wearable; haptic; linear resonator actuator; pulse width modulation; vibratory sensation; grinding force





## 1. Introduction

Over the past few decades, technological advancements have been staggering, with flourishing research in areas such as communications, actuators, sensors, remote control, and monitoring. The integration of these fields has fostered the development of industries such as manufacturing, entertainment, education, and more, providing the necessary technology for various wearable products and thereby enhancing the quality of human life.

According to the latest market research analysis on wearable devices, North America, the largest market for wearables in 2022, accounted for 33.8% of the global market share, with the global market reaching a total of USD 61.3 billion in that year [1,2]. The increasing demand in various regions worldwide suggests that wearables are becoming increasingly indispensable for humans, indicating that these types of products have the potential to transform human lifestyles on a large scale.

Currently, the market for wearable devices is dominated by hand and head wearables, with a notable example being the Apple Watch, produced by Apple Inc. This smartwatch combines the functionality of health monitoring and recording, providing comprehensive tracking of users' physical health and integrating various indicators. This has made it possible to measure certain parameters in real-time and provide personalized diagnosis, tasks that used to require hospital visits. Moreover, the device can provide different services based on the user's needs, such as child location monitoring and elderly sleep monitoring. As for the future development of this field from 2023 to 2030, related research predicts a Compound Annual Growth Rate (CAGR) of 14.6%, with sales forecast to reach approximately USD 186 billion by 2030 [2–4].

Interaction between wearable devices and humans is an interesting research area, encompassing aspects such as wearable positioning [5], applicable scenarios [6,7], and system design [8]. Most devices are operated via touch, but when it comes to feedback on human senses, vision and hearing are primarily used. This is because vision and hearing are the two most valued senses by humans [9], hence the market demand and requirements for products extending from these two senses are relatively high. In addition, the tactile sense, which is the most widely distributed sensory system in humans, plays a crucial role in interactions with the outside world [10]. There are many types of tactile receptors, each with different characteristics [11,12], which is why artificial tactile systems are currently limited in their ability to convey actual physical quantities of touch [13].

The most common tactile application in today's market is the vibration generated by a device to alert users. Apple Inc. developed the Taptic Engine, based on the Linear Resonator Actuator (LRA), to simulate the feedback sensation of pressing a button, aiming to enhance the user experience. Moreover, research on tactile sensing is advancing on various fronts globally. For example, Sim et al. have developed a thin film element and its signal interpretation method that can detect heat and pressure simultaneously, leveraging the nanowire of ZnO and the Seebeck effect to convert physical quantities into electrical signals to detect different external stimuli [14]. Youn et al. have developed a soft Dielectric Elastomer Actuator (DEA) that demonstrates high actuation force over a wide operating frequency range, with the maximum output force reaching 8.48 N [15]. The concept of a "Tactile Avatar" proposed by Kim's group combines an array of piezoelectric sensors with a deep learning network that simulates the human tactile learning process, enabling it to react to the roughness and hardness of different materials similarly to humans [11]. These studies are just the tip of the iceberg, indicating that there is enormous potential for development in research related to human tactile senses.

This paper introduces a tactile wearable device that integrates a three-axis commercially available LRA and MEMS accelerometer. The primary function of this device is to produce a vibratory sensation on the human finger and detect the feedback acceleration signals from the finger. The grinding force signal from the machining process can be transmitted to the host of the tactile wearable device, the Raspberry Pi 4b, via the internet or in the form of recorded files. The host converts the vibrational force into PWM commands, which are then sent to the device's driver through USB. The driver interprets the PWM commands, generating actual PWM signals to control the LRA driver. The AC signal produced by the LRA driver induces vibrations in the LRA, which are then captured by the accelerometer and sent back to the Raspberry Pi for storage and subsequent data analysis.

This paper is organized as below: The second part of this paper will introduce the relevant background knowledge, covering the importance of grinding force produced by grinding machines during the machining process, the principles of human skin perception of mechanical stimuli (such as vibration and pressure), and the basic characteristics of LRA.

The third part details the creation of the self-developed wearable haptic system for subsequently vibratory sensation, including its characteristics, hardware architecture, software architecture, and other equipment.

The fourth section describes the experimental process to acquire the grinding force signals measured from the actual machine (grinding machine).

The fifth section covers (1) the frequency domain analysis of the acquired grinding force signal; (2) the input of the measured remote grinding force into the Raspberry; and (3) the algorithms for converting the grinding force signal into a PWM duty cycle. (4) The internal control system generates corresponding PWM signals for the LRA driver based on these values, and (5) the acceleration is captured through the accelerometer, and the data are returned and stored in the Raspberry. (6) Discussion on the time-domain and frequency-domain of the LRA-driven signals and acceleration signals.

The last part presents the conclusions drawn from this experiment and potential areas for improvement.

## 2. Background Study

### 2.1. Monitoring Grinding Force

Grinding is one of the most widely used, high-efficiency, low-cost finishing techniques in production systems. The grinding force causes the machined surface to be scratched, ploughed, and sliced by abrasive grits during the grinding process due to the high-speed mutual movement between the grinding wheel and the workpiece [16].

Grinding force is a critical quantity for determining grinding performance since it has a direct influence on wheel longevity, grinding specific energy, heat source temperature field, workpiece surface quality, and shape-position precision [17].

Monitoring precision grinding processes improves knowledge of the complicated material removal process, enables workpiece productivity and quality improvement, and reduces production costs [17]. Grinding force could be used to detect grinding wheel wear, calculate energy, minimize chatter, adjust force, and simulate the grinding process. The grinding force could also be used to adjust the machining settings as well as the construction of the grinding machine and fixture in order to fully utilize the grinding process's potential [18].

Currently, commercial products are available to help users select the right grinding setting [19]. By visualizing the grinding process in terms of process forces, spindle motor power consumption, and elastic deformation or feed motions of the grinding machine's axis, the grinding process could be evaluated and optimized.

Nowadays, robots have significantly aided in automating repetitive tasks for which position control-based solutions are effective. However, machining activities needing both position and force control, such as grinding or deburring, are still carried out manually by expert operators, mostly due to the lengthy and inflexible nature of programming, which prevents them from being transferred to robotic solutions. A hand-held instrumented tool capable of capturing 3D interaction forces and torques between the hand-held tool and workpiece was reported [20].

A method for instrumenting hand-held grinding tools for measuring the performance of trained human operators was created and is intended for future robot programming and control [21]. Their instrumented tool is intended to measure interaction forces between the workpiece and the contact tool.

Therefore, in this study, we monitor the interaction force between the grinding wheel and the workpiece held by the operator and remotely transmit the force to a wearable system to facilitate the apprentice or the robot programmer feeling the grinding force with vibratory sensation.

### 2.2. Haptic Mechanisms in Humans

The skin is the largest organ of the human body, and mammals have two types of skin: glabrous (hairless) and hairy. Glabrous skin can be used to recognize the shape and material of objects, while hairy skin is more related to emotional touch [22]. Underneath the skin, there are many types of sensory structures that help humans discern external stimuli such as temperature and touch. These sensory structures' receptors mainly consist of mechanoreceptors, thermoreceptors, and pain receptors. This paper primarily deals with the employment of mechanical forces exerted on human skin; consequently, our review in this section emphasizes the study of mechanoreceptors.

External mechanical stimuli depend on low-threshold mechanoreceptors (LTMRs) and related nerve conduction to the brain [23]. Based on function and adaptation speed, LTMRs can be categorized into fast-adapting (FA or RA) and slowly adapting (SA) categories. According to morphology, they can correspond to four sensory structures: Merkel cell-neurite complexes (SA I), Meissner corpuscles (FA I), Ruffini corpuscles (SA II), and Pacinian corpuscles (FA II) [12,24].

Merkel cell-neurite complexes exist in both hairy and glabrous skin, mostly in the human epidermis [24]. They belong to SA I and therefore have a higher spatial resolution, playing a crucial role in regulating gentle touch. Recent research has concluded that Merkel

cells are touch-sensitive cells that convert mechanical stimuli into irregular, continuous electrical signals through the Piezo2 protein [25]. Meissner corpuscles are located at the same depth in the skin as Merkel cells, mainly in glabrous skin, especially on the fingertips [26]. They are responsible for transmitting low-frequency vibration signals (10~50 Hz) and the sliding touch feeling between the object and the skin. The receptive diameter range of a single receptor is about 3~5 mm, and it responds uniformly to stimuli within this range, hence a lower resolution [27,28]. Ruffini corpuscles, located in both the hairy and glabrous dermis, are thought to be used for skin stretch detection. The density of Ruffini corpuscles in human fingers is estimated at <0.3/mm² [29], so the relevance of Ruffini corpuscles categorized as SA II remains debatable [12]. Pacinian corpuscles are indispensable for high-frequency vibrational touch and are located in the glabrous dermis, among other locations depending on species [22]. Their unique onion-like capsule acts as a filter, allowing only high-frequency mechanical signals to stimulate nerves [24]. For electrical stimulation simulations of Pacinian corpuscles, the threshold reaches a minimum at around 200 Hz [30]. The Table 1 below summarizes the key features of the four types of LTMRs.

**Table 1.** Characteristics table of LTMRs.

|  | Merkel Cell-Neurite Complexes | Meissner Corpuscles | Ruffini Corpuscles | Pacinian Corpuscles |
|---|---|---|---|---|
| LTMR types [12,22] | Aβ SA I | Aβ FA I | Aβ SA II [1] | Aβ FA II |
| Sensitive frequency (Hz) | 5~15 [31] | 10~50 [28] | 0~8 [32] [2] | 200~400 [12] |
| Ratio (hand) [27] [3] | 25% | 40% | 20% | 10–15% |
| Location [33] | Tip of epidermal sweat ridges | Dermal papillae | Dermis | Dermis (deep tissue) |
| Best stimulus [33] | Edges, points | Lateral motion and low frequency vibration | Skin stretch | High frequency vibration |

[1] From the perspective of histological analysis, its density is quite low, so some people remain skeptical about this association. [2] Only the range has been found. [3] It is an inferred result from electrophysiology, not an actual proportion from anatomy.

Human touch is the result of the regulation and coupling of the above four types of LTMRs with other nerves. Therefore, the touch threshold will vary with frequency. Research by Brisben's group indicates that when humans grip with their hands, the lowest (most sensitive) perception threshold corresponds to vibrations between 150 Hz and 200 Hz [33]. Therefore, many tactile actuators have inherent resonant frequencies within this range, including the actuator-LRA used in this study.

### 2.3. Characteristics of the LRA

The Linear Resonant Actuator (LRA) is a small brushless motor capable of generating vibration. It is commonly used in consumer electronic products such as mobile phones, game controllers, and haptic feedback devices to provide tactile or vibration feedback. The structure of an LRA consists of a combination of a metal spring and a magnet assembly. When an alternating current is applied, the magnet generates a driving force on the metal spring, causing the LRA to vibrate. The frequency and amplitude of the LRA's vibration can be controlled by adjusting the frequency and amplitude of the current. This technology can generate highly precise vibration feedback and is widely used due to its low power consumption and rapid response.

LRAs can be classified into single-axis degrees of freedom and multi-axis degrees of freedom [34]. Many studies have discussed the applications and control methods of multi-axis degree of freedom LRAs [35,36]. An LRA is a mass-spring system and has a fixed resonant frequency ($f_r$) once designed. The primary design range of $f_r$ falls between

150 and 200 Hz, and the input frequency of the working voltage should be referred to $f_r$ to maximize the actuation force. The resonant frequency can be written as:

$$f_r = \frac{1}{2\pi}\sqrt{\frac{k}{m}} \tag{1}$$

where $k$ represents the system's rigidity and m is the system's mass. Notably, the resonant frequency may shift due to external loads, such as user pressure or installation method. Therefore, some studies aim to explore how to track the offset of the LRA's resonant frequency to improve operational efficiency. The commonly used method is Back Electromotive Force (Back-EMF, BEMF) sensing [37,38]. Back-EMF can be described by Faraday's law in terms of its magnitude:

$$\varepsilon = -\frac{\partial \Phi_B}{\partial t} \tag{2}$$

where $\varepsilon$ is the electromotive force magnitude, and $\Phi_B$ is the magnetic flux. When the LRA operates at $f_r$, BEMF will also reach its maximum value. Therefore, it is possible to extract BEMF from the signal output to the LRA through a special circuit, and by observing the change in BEMF combined with the least mean squares method, real-time tracking of the resonant frequency can be achieved. Currently, companies like Texas Instruments (TI) have already demonstrated the feasibility of this tracking algorithm using specific driver chips (e.g., DRV2605L) [38].

## 3. Wearable Haptic System

### 3.1. Functional Block Diagram of a Wearable Haptic System

A novel three-axis wearable haptic system was developed, named the Remote-Controlled Wearable System (RCWS). It was divided into three major sections according to functionality: the RCWS server, the RCWS driver, and the vibratory ring. The RCWS server is responsible for receiving machining force information and user input information and converting the information into RCWS driver commands according to a specific format (more details are described in Section 3.3.2); the RCWS driver is responsible for the control of LRA and the capture of acceleration signals; and the vibratory ring provides vibrational tactile feedback to the operator's finger.

The main core of the RCWS server is a Raspberry 4b, which is primarily tasked with receiving real-time or stored grinding force data and converting the grinding force into Pulse Width Modulation (PWM) commands. These PWM commands are transmitted to the RCWS driver via a USB cable, where they are interpreted by the MCU (Microcontroller Unit: STM32F412RET6) and generate actual PWM signals to control the LRA (VG1040003D) driver. The AC drive signal emitted from the LRA driver is applied to the LRA, causing it to vibrate. The RCWS driver has a total of three LRA drivers, which are orthogonal to each other, thus enabling the three LRAs on the vibratory ring to have different vibration magnitudes. Simultaneously with the vibration, the accelerometer (ADXL355) on the vibratory ring reports the three-axis acceleration to the MCU at 4000 Hz. The MCU buffers the acceleration data and sends it back to the RCWS server to be saved as a file when a fixed quantity of data isreached. The power for the RCWS driver and vibratory ring is supplied by the Raspberry's USB, drawing an average current of around 400 mA during operation. The hardware selection and software architecture of the RCWS will be detailed in the remaining parts of this section. Figure 1a illustrates the system architecture of RCWS. Subsequently, part (b) of Figure 1 represents the forward force of grinding measured through a load cell (Section 4), while part (c) depicts the machining vibrations simulated by RCWS, allowing users to experience the vibrations generated during remote machining.

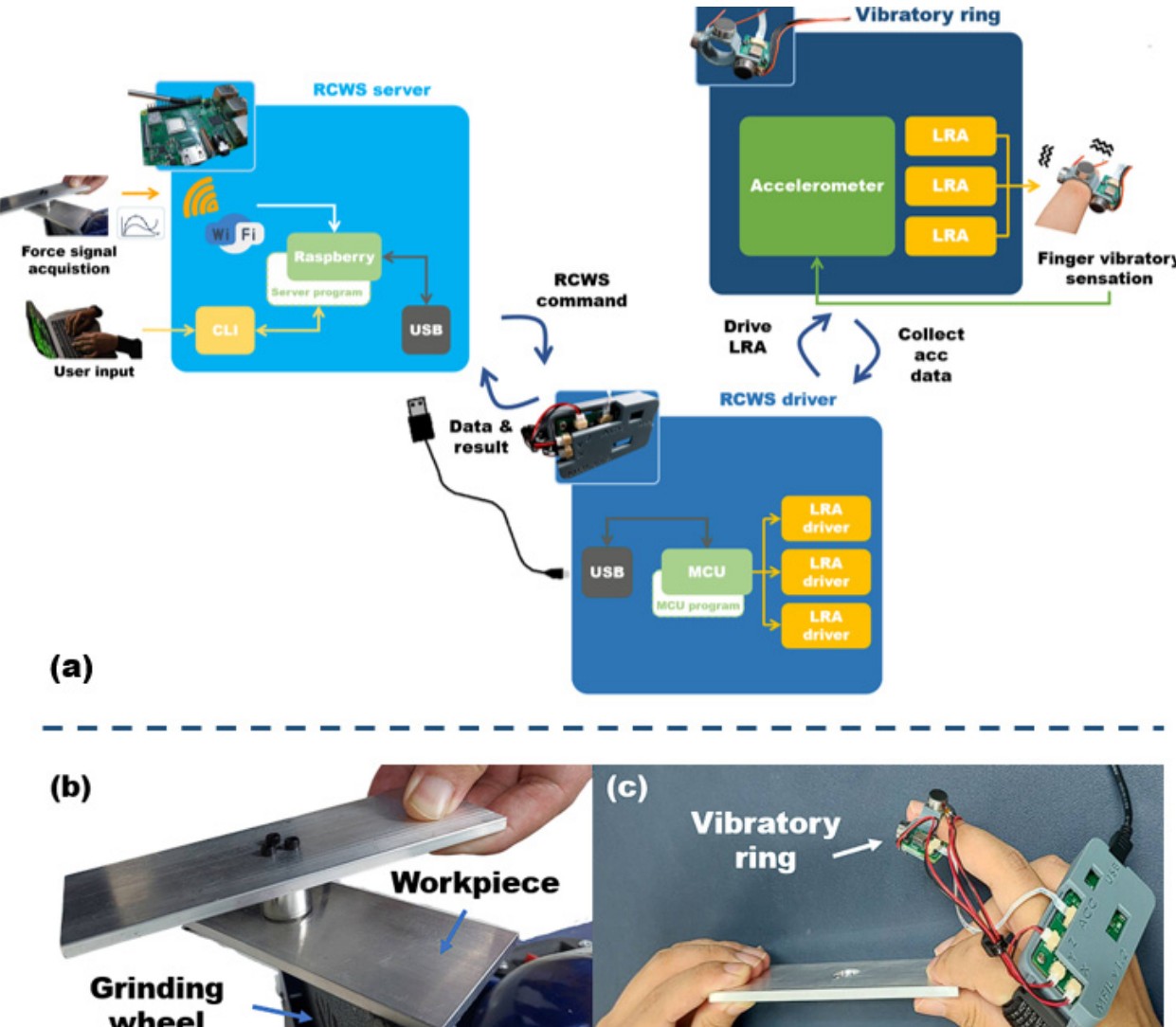

**Figure 1.** (**a**) System architecture of RCWS; (**b**) Measurement of grinding force through the load cell system described in Section 4; (**c**) Vibrations generated during remote machining experienced by the user wearing RCWS.

### 3.2. RCWS Hardware

### 3.2.1. MCU-STM32F412RET6

MCUs often serve as the main controllers for microdevices with real-time control requirements. STMicroelectronics' STM32 is a standout in 32-bit MCUs, with a maximum clock speed of 100 MHz and incorporating common communication protocols for external devices (e.g., I2C, SPI, and PWM). Additionally, STM32 features a powerful timer that can achieve microsecond-level timing by adjusting internal dividers, and its interrupt system allows the controller to obtain primary data when external device data updates. The microprocessor can be divided into the F, H, and G series, among others, depending on features and performance. The RCWS uses the most common F4 series as the controller, performing data read/write operations on external devices centered around this unit. Simultaneously, the F4 series also provides a USB 2.0 communication interface, allowing other devices to transmit data through a USB hub, and the RCWS server interacts with the controller on the driver board using this protocol.

### 3.2.2. Accelerometer-ADXL355

The ADXL355 is a highly accurate MEMS digital accelerometer with a data bit accuracy of 20 bits and three measurement ranges of 2/4/8 g. The internal data collection process starts with an analog signal passing through a sinc3 low-pass filter. This filter has a −6 dB attenuation at 1.5 kHz. The signal is then converted into a digital signal via a 20-bit precision Σ-Δ ADC. The acquired signal goes through programmable digital low-pass and high-pass filters, and the latest data is finally placed in corresponding registers. The ADXL355 utilizes a FIFO (First In, First Out) data buffering mechanism, allowing users to access past data records by reading the contents of the FIFO buffer.

The ADXL355 has two additional mechanisms that work in synergy with the MCU. The first is the interrupt pin DRDY, which transitions to high when data is updated. Paired with the MCU's interrupt mechanism, this can ensure that the MCU receives the latest data at the right time. The second mechanism is the SPI multiple-byte read/write ability, which allows the MCU to perform IO operations on the accelerometer register at approximately 5 Mbits/s.

Within the RCWS, ADXL355 serves to capture the vibrational signals simulated by the LRA actuator, providing data to the MCU for further transmission to the RCWS server.

### 3.2.3. Actuator-VG1040003D (LRA)

The VG1040003D is a small LRA, with a diameter of just 10 mm and a height of only 4 mm. Its compact size positively impacts the reduction in size of the part of the RCWS worn by the user. Wiring LRA is simple and can be soldered directly onto the contact pad using AWG 28 wire. This LRA can generate an acceleration of approximately 2 Grms (2.5 Vrms, AC), with a maximum rise time of 10 ms and a fall time of around 50 ms. The measurement time is based on the difference between the absence of vibration and the target vibration size. Importantly, this LRA offers a wider bandwidth of 150 to 200 Hz, in contrast to products like Precision Microdrive's model No. C10-100, which is confined to a range of 170 to 180 Hz. The term bandwidth refers to the frequency range where acceleration can reach at least half of its peak value [39].

### 3.2.4. LRA Driver–DRV2605L

The DRV2605L is an integrated LRA and ERM (Eccentric Rotating Mass) drive circuit driver IC that requires a minimum power of 2 V for operation. This driver offers diverse control methods, including PWM, RTP (Real-Time Playback), and even analog audio drives. In RCWS, the output of the DRV2605L is controlled in an open-loop mode by PWM, with two adjustable parameters: duty cycle and frequency (the amplitude of PWM is fixed). The former parameter influences the steady-state peak magnitude of the output, and Figure 2a illustrates the relationship between the duty cycle and the peak voltage output. The peak voltage can be adjusted through the OD_CLAMP register, which is set to 140 in this experiment, corresponding to 2.97 V. The steady-state peaks of the positive and negative values are phase-shifted by 180 degrees. The latter parameter, frequency, is directly proportional to the output voltage frequency; dividing the PWM frequency by 128 yields the frequency used to drive the LRA. As the maximum amplitude of VG1040003D occurs at 170 Hz, the driving PWM frequency is consistently set at 21,760 Hz in this study. Figure 2b describes the relationship between the PWM's duty cycle and frequency and the schematic output signal.

The DRV2605L uses I2C as its communication interface. I2C passes messages through addressing, and since each DRV2605L has the same address, an I2C mux is needed as an intermediary layer.

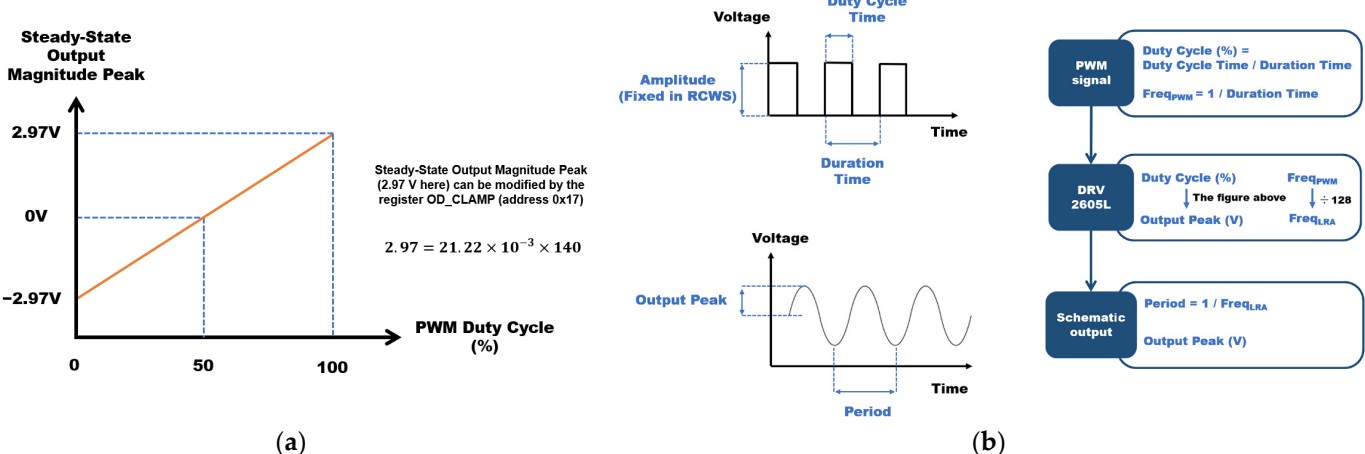

**Figure 2.** (**a**) Duty cycle versus peak of output voltage diagram; (**b**) schematic diagram of the PWM signal converted to the output signal through DRV2605L.

### 3.2.5. LRA Driver–DRV2605L

The STM32 is focused on providing flexible integration with sensors or drivers. However, it lacks built-in network transmission capabilities, so a device with a wireless or wired network is needed as an intermediary layer, allowing the wearable device to connect with the Internet. One suitable option that meets network function is a Linux-based microdevice, such as the Raspberry series, which supports Wi-Fi connections in the 2.4 G and 5 G bands. In this work, Raspberry 4b fulfills this requirement. It features a USB hub that can connect with the RCWS controller, allowing control of the internal status of the RCWS through the command-line interface (CLI).

### 3.2.6. Wearable Hardware Implementation of RCWS

To prevent direct contact between the circuit, or LRA, and the human body, 3D-printed photopolymerization technology is used to produce wearable mechanisms. The wearable mechanisms include two parts: the vibratory ring (Figure 3a) and the RCWS driver (Figure 3b). The vibratory ring that can be worn on the human finger was designed. This ring is assembled with a MEMS accelerometer (Figure 3c) and three orthogonally arranged LRAs. The ring has three flat plates that are perpendicular to each other for securing the LRA. The LRAs were attached to these flat plates using epoxy resin. The accelerometer is then secured to the 3D-printed structure using a Ø 1.7-mm bolt. This single accelerometer was mounted on the 3D-printed plate perpendicular to the Z direction so that it could detect the X-Y-Z axis acceleration produced by LRAs.

The RCWS driver board (Figure 3d) is packaged in a square box that can be disassembled from the middle assembly interface, and the lugs on both sides of the box secure the watch pin. The finished square box with the driver board is worn on the user's hand through the watch strap (Figure 3e,f).

### 3.3. RCWS Software

#### 3.3.1. RCWS Software Flow Chart

The RCWS software can be divided into two major blocks by USB: one is the RCWS server dealing with user input and related programs of vibration data, and the other is the program on the driver board controlling LRA, capturing acceleration, transmitting acceleration, and parsing and executing commands. Figure 4 is the overall software architecture diagram of RCWS, which describes the processing logic of each execution sequence or interruption in the program from a global perspective.

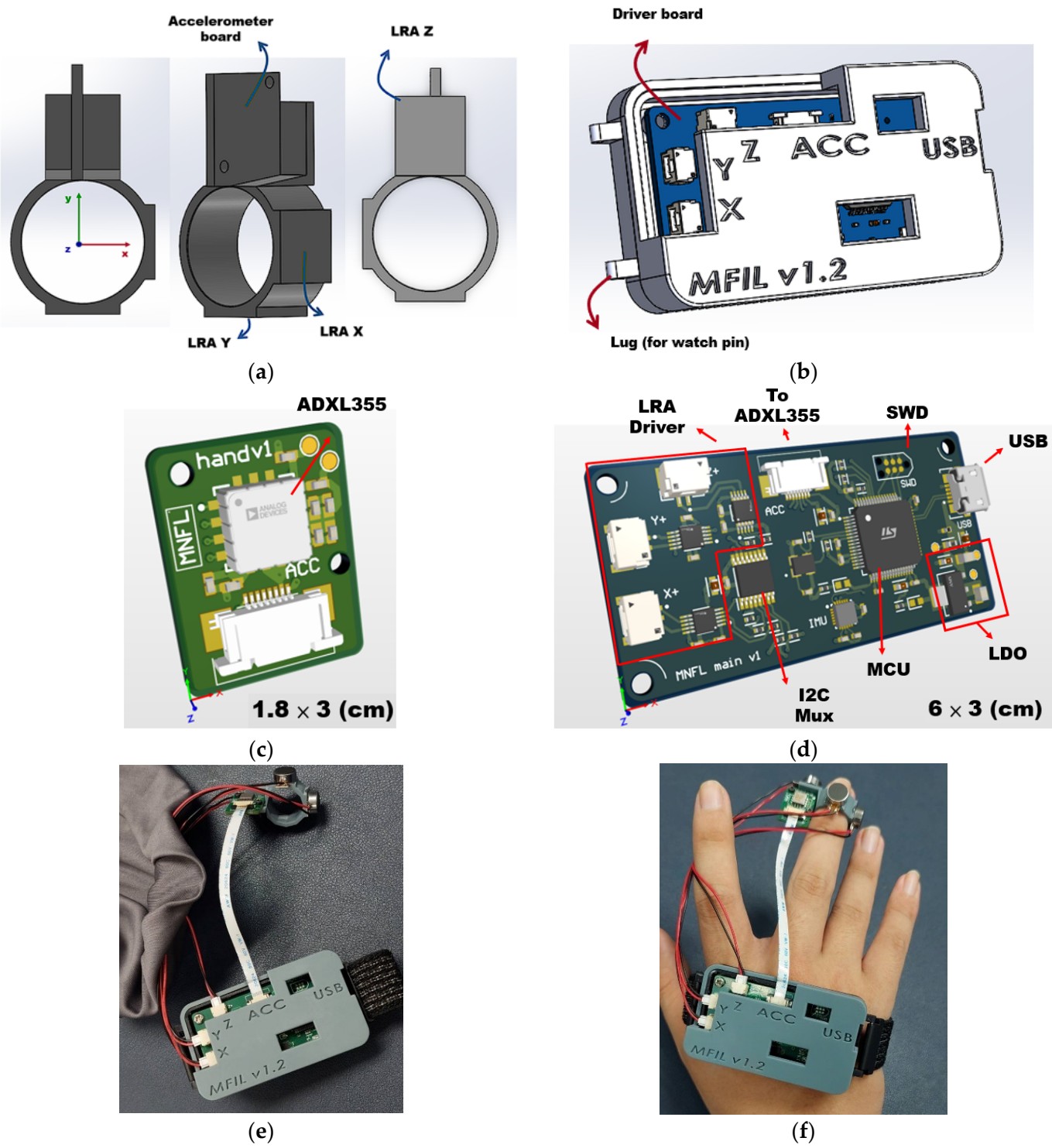

**Figure 3.** (**a**) RCWS vibratory ring wearable on the user's middle or index finger; (**b**) RCWS driver housing with lugs on both sides for attaching the strap; (**c**) PCB layout of the RCWS accelerometer; (**d**) PCB layout of the RCWS driver; (**e**) assembly diagram of the RCWS; (**f**) user wearing the physical model of the RCWS.

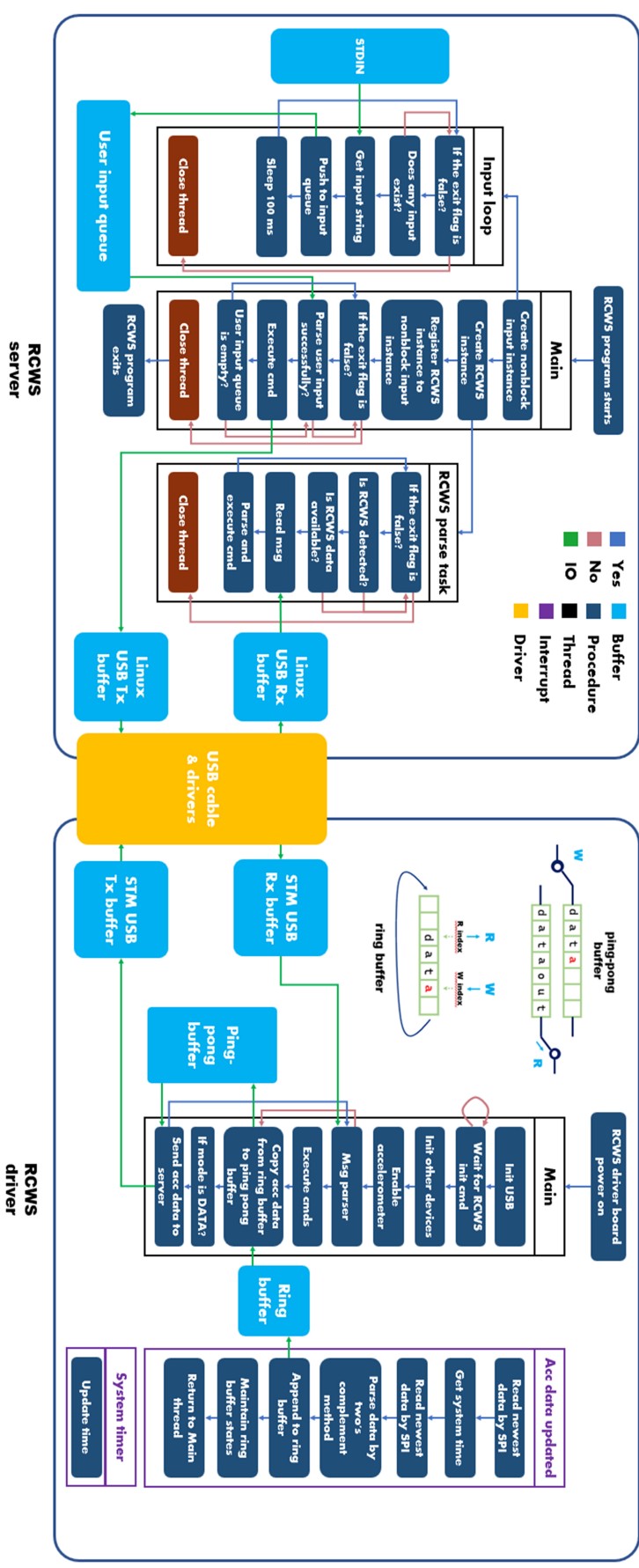

**Figure 4.** Software architecture diagram of RCWS.

3.3.2. Definition of the Message Structure of USB Transmission

To ensure the proper construction and parsing of messages between the RCWS server and the driver, a unified message structure must be defined.

The name of this structure is the "RCWS USB packet frame", and all commands (see Section 3.3.3), such as the PWM command, acceleration, etc., must be converted into this structure before being transmitted via USB. In USB communication, the terms OUT and IN represent directions, with OUT signifying the data flow from host to function and IN signifying the data flow from function to host. In this context, the server acts as the host, and the driver serves as the function. Figure 5 illustrates the composition of this structure, which is composed of four parts: Cmd_Byte, Data_Len, Data, and End of Packet (EOP).

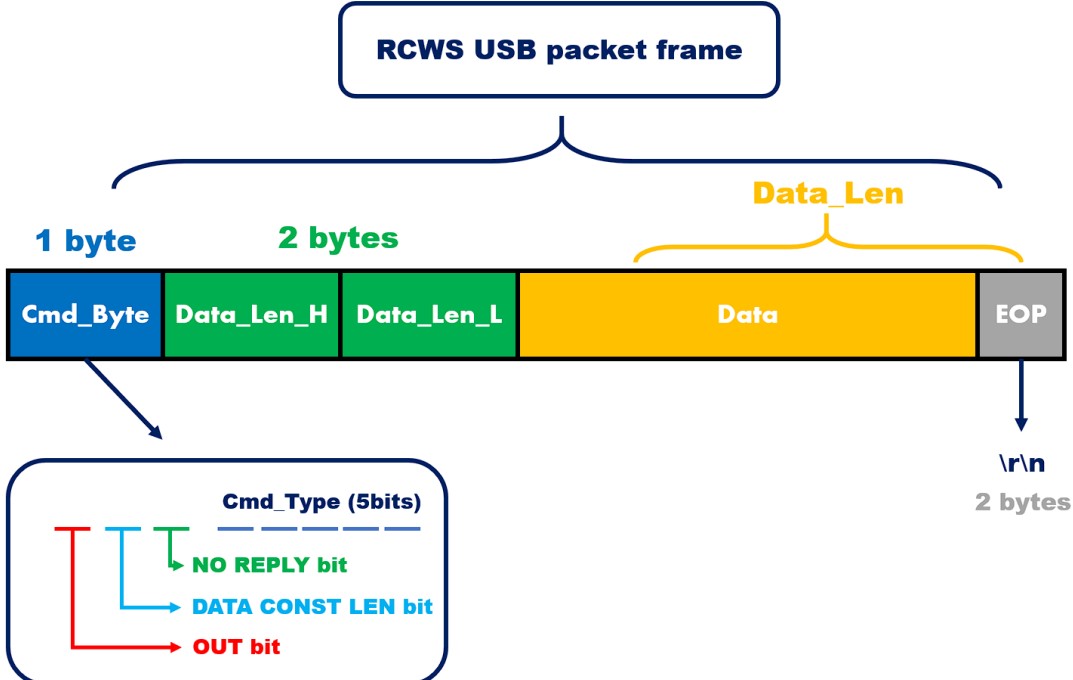

**Figure 5.** The RCWS USB packet frame is composed of four parts: Cmd_Byte, Data_Len, Data, and EOP.

Cmd_Byte is 1 byte (8 bits) in length, and it is divided into segments of 1, 1, 1, and 5 bits in length from the most significant bit to the least significant bit. This allows for the inclusion of four types of information, representing, in descending order of significance: the direction of message transmission, whether the cmd_type should be of constant length (DATA_CONST_LEN), whether the receiving end needs to reply, and the type of the message. When the DATA_CONST_LEN bit is set, it enables pre-checking the length for messages that are always of equal length, allowing for preliminary validation of the message before parsing.

Data_Len is determined by the sum of the byte counts of Data and EOP, consisting of two bytes, so the longest message can reach 65,535. Data is the primary storage location for the information to be transmitted, and it has different formats depending on the Cmd_Byte. The parsers at both ends interpret the content of Data according to cmd_type. EOP is defined as "\r\n," indicating the end of the message.

The RCWS USB packet frame is ultimately converted into USB bulk packets (64 bytes per bulk packet) by the underlying USB driver. At the other end, the packets are reassembled by the corresponding underlying USB driver.

Table 2 displays all command indices, with the note that some of them are not available for the user. Among them, the commands highlighted with an orange background are used for system-level information transmission.

**Table 2.** RCWS commands table: The commands highlighted with an orange background are designated solely for system usage. The details of these commands are described in Section 3.3.3.

| Cmd_Type Index | Cmd Name | Data_Len | User Available | Direction |
|---|---|---|---|---|
| 1 | System log | Not const | X | IN |
| 2 | Open | 3 | V | OUT |
| 3 | Switch mode | 3 | V | OUT |
| 4 | Init | 21/21 | V | IN/OUT |
| 5 | Update register | Not const | X | OUT |
| 6 | Get register | Not const/5 | V | IN/OUT |
| 7 | Reset IC | 3 | V | OUT |
| 8 | Parse error | Not const | X | IN |
| 9 | PWM command | 36/32 | V | IN/OUT |
| 10 | Acceleration | Not const (see Section 5.5) | X | IN |

Among the many commands, Acceleration, and PWM command are the most critical, and their data details are described below.

From the raw data extracted from the accelerometer, the data length for each axis is 3 bytes. Based on the current measurement range of the accelerometer, the data is parsed using two's complements. The parsed three-axis data, along with the capture time, form an acceleration data structure. Figure 6a illustrates an example of the acceleration parsing process, while the left half of part (b) in Figure 6 shows the storage structure of the data after parsing.

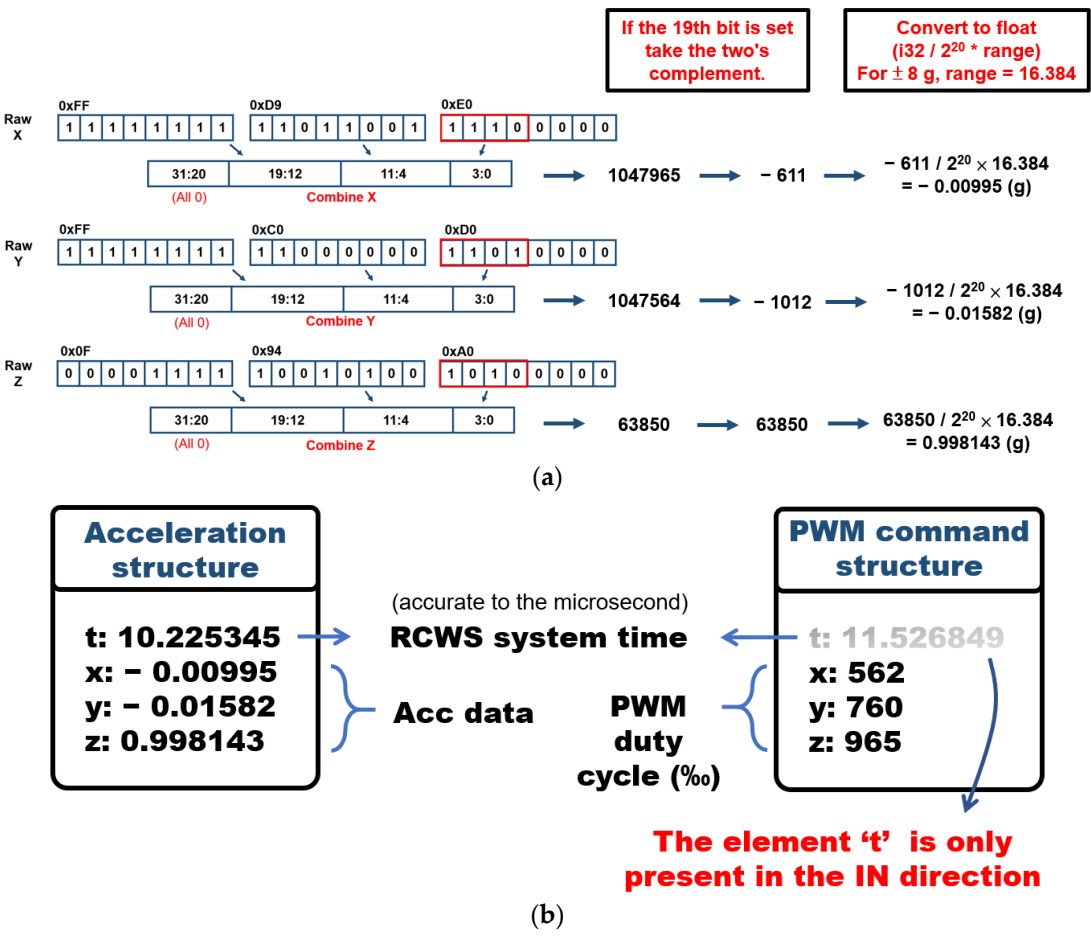

**Figure 6.** (**a**) The data parsing process when the accelerometer is laid flat; (**b**) the two most commonly used data structures in RCWS.

The PWM command includes the PWM duty cycle for all three axes. The right half of Figure 6b represents the data structure of the PWM command. The time information is included only in the PWM command sent back from the driver's side. The purpose of this is to inform the server of the modified PWM signal and the time of modification, allowing the server to record the changes. This facilitates subsequent data analysis by ensuring that the time axes of the PWM information and acceleration information are aligned.

### 3.3.3. RCWS Driver Modes and CLI

The driver in RCWS operates as a finite-state machine, i.e., the software design pattern where a given model transitions to other behavioral states through external input, having four distinct modes: NONE, WAIT_FOR_INIT, CTRL, and DATA.

- NONE: This is the state when USB communication between the driver and RCWS server has not yet been established.
- WAIT_FOR_INIT: This state occurs after USB communication has been established but before receiving the initialization message.
- CTRL: This state is entered after receiving the initialization message, allowing for operations such as resetting and reading the internal IC. The vibration ring is at rest in this state, and vibration generation is prohibited.
- DATA: This state activates the vibration ring, and acceleration signals start being sent back to the server. CTRL and DATA modes can be switched through the "switch mode" command.

The standard startup process is as follows, and the commands used will be detailed in the CLI section:

1. When the driver is first started, it initializes the mode to NONE and waits for USB communication to be established.
2. After the server-side uses the "open" command, the mode transitions to WAIT_FOR_INIT.
3. Once the server-side uses the "init" command to send an initialization string that is verified as correct, the state switches to CTRL.
4. When the user wants to activate the vibration ring, the "switch mode" command must be used to change the state to DATA.

This process is shown on the left side of Figure 7a, while the right side lists the command types that the driver allows to be send/receive in CTRL/DATA mode. All commands from the server are first classified by type after being parsed. Moreover, depending on the current mode of the driver, they are either further parsed or ignored.

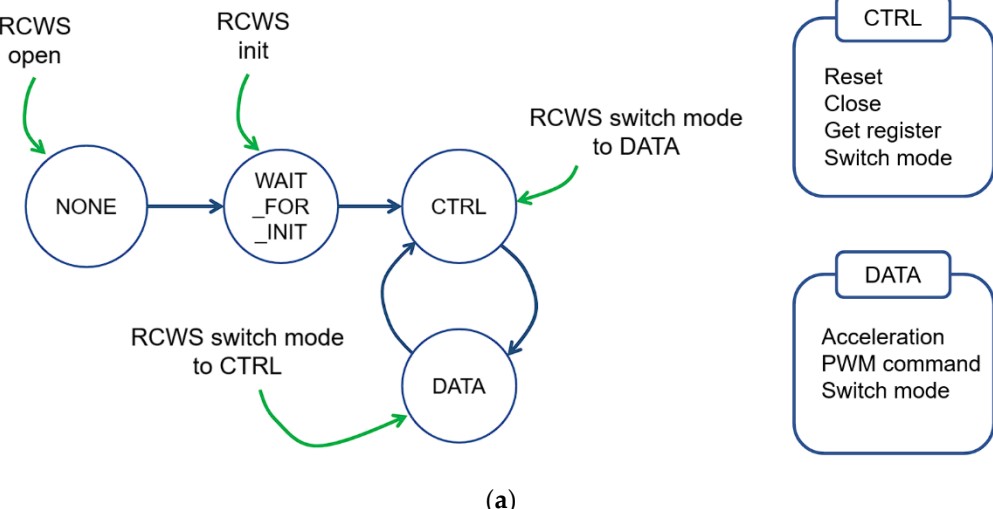

(a)

**Figure 7.** *Cont.*

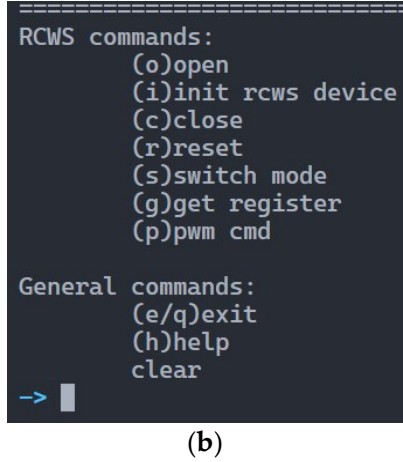

(b)

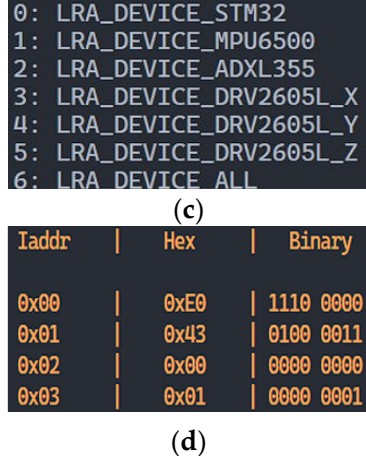

(c)

(d)

**Figure 7.** (**a**) RCWS driver startup process diagram and allowed commands in different modes; (**b**) RCWS CLI; (**c**) This figure presents the IC index for all ICs located on the driver board; (**d**) The results of reading part of the DRV2605L registers.

The RCWS server provides several commands through the CLI for users, which are described in more detail below. Figure 7b is a screenshot of the CLI.

- Open: Upon the initiation of this command, the RCWS server conducts a comprehensive exploration of all available USB ports to identify and enumerate RCWS driver instances. If multiple RCWS drivers are detected concurrently, the system affords the user the capability to select one from among the alternatives. After the selection of a specific driver, a USB communication channel is created, and the corresponding driver is informed of the successful establishment of a connection with the server. Concurrently, the mode of the driver undergoes a transition to the WAIT_FOR_INIT state.
- Close: This command disconnects the server from the driver.
- Init: This command transmits a designated initialization string, thereby signaling the RCWS driver to enter the initialization phase. Upon successful validation of the initialization string, the driver mode is consequently transitioned to the CTRL state.
- Reset: All ICs on the driver can be reset to their factory settings. This command enumerates all the ICs on the driver, allowing users to selectively reset the ICs of their choice to their original configurations. Figure 7c, the left image, shows the index of the ICs on the driver.
- Get register: This command allows users to read the register values from a specific IC. The first parameter is the IC index, the second parameter is the start address, and the third parameter is the end address. Addresses can be expressed in either decimal or hexadecimal notation. The results of the reading will be displayed on the screen in both binary and hexadecimal formats. The right image in Figure 7d shows the results of reading part of the DRV2605L registers.
- Acceleration: This command is actively called by the driver in DATA mode (not available for the user). When the amount or length of the accumulated acceleration data reaches a certain quantity (details in Section 5.5), the driver will execute this command. The buffered acceleration data will be transmitted to the server and recorded in a txt file.
- PWM command: The PWM duty cycles for the three axes are packaged together and sent to the driver. The duty cycles are expressed in permille, so for example, if you want a PWM duty cycle of 50%, it would be marked as 500. The driver will adjust the PWM signals output by the MCU according to this command.
- Switch mode: Users can change the driver mode with this command.

## 4. Force Measurement in a Grinding Process

### 4.1. Grinder and Workpiece

The experimental setup for obtaining the grinding force signal in this study employed a bilateral 8-inch grinding wheel machine, the MD-100. Under the rated voltage of 110 V, the full-speed rotation speed is 3450 rpm. The configured grinding wheel size is 200 × 20 × 16 mm. The weight of this machine is approximately 9.2 kg. This machine operates through a single switch and requires about 40 s to reach maximum speed without the function of speed adjustment. Figure 8a is a side view of the grinding machine used in this study.

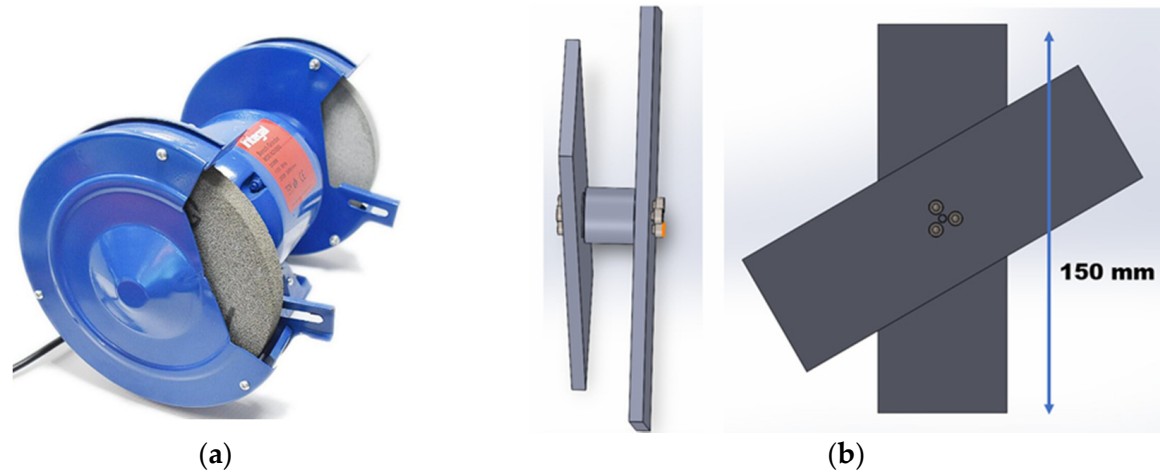

**Figure 8.** (**a**) Exterior view of the MD-100; (**b**) Side and front views of the workpiece.

The workpiece to be ground by the grinding wheel machine was a flat aluminum alloy plate made of Al 6061. The external dimensions of the aluminum alloy plate are 150 mm × 50 mm × 5 mm. The entire home-made workpiece is composed of two aluminum flat plates and a load cell fastened together with bolts, as shown in Figure 8b.

### 4.2. Load Cell, DAQ, and Calibration

The load cell (GIS-FA404) was used to measure the grinding forces. This load cell was constructed with a strain gauge, primarily measuring tensile/compressive forces. When its structure is subjected to an external force causing strain, the resistances of the four internal strain gauges will correspondingly change, causing a slight voltage change. Figure 9a illustrates the schematic diagram of the internal circuitry of the GIS-FA404.

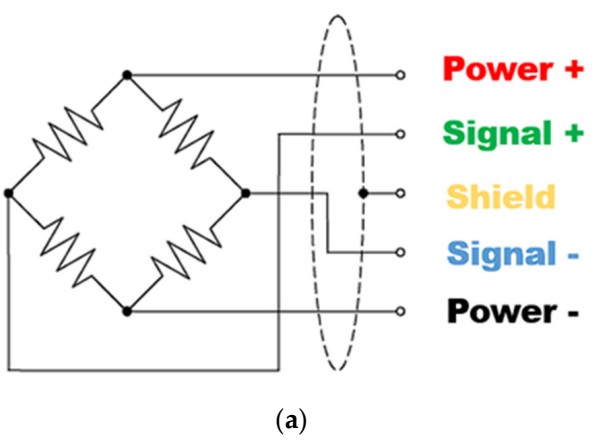

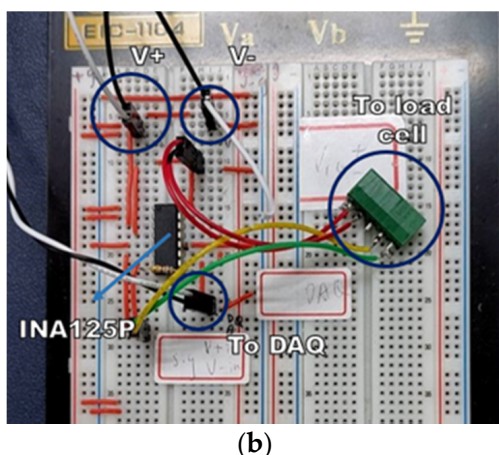

**Figure 9.** *Cont.*

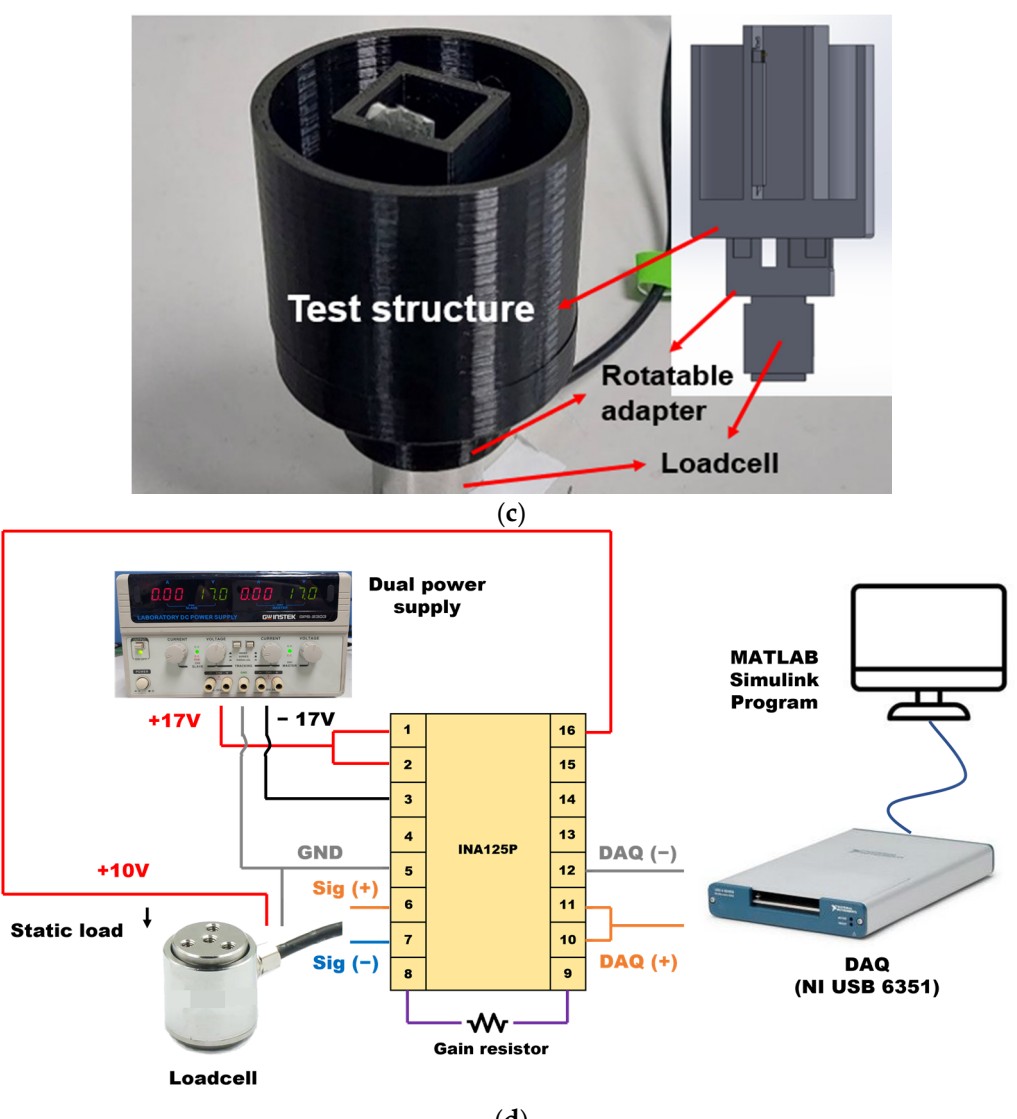

**Figure 9.** (**a**) Schematic diagram of load cell internal circuit; (**b**) physical circuit diagram of INA125P; (**c**) structure fixed on the load cell for placing weights; (**d**) setup diagram for measuring load cell linearity.

The GIS-FA404 can measure up to 20 N of external force. The output sensitivity provided by the manufacturer is 2.325 mV/V, based on the power input voltage. Even when the maximum input voltage of 15 V is applied, the full-scale output will still be greatly affected by environmental noise. Therefore, a differential voltage amplifier needs to be placed at the back end of the load cell. The INA125P voltage amplifier is a very suitable option. The INA125P has several distinct advantages. The first is that it can provide a fixed voltage of 10 V for the load cell power, so an additional power supply for the load cell is not required. The second advantage is that this amplifier has an adjustable amplification ratio, which can achieve the desired amplification effect by replacing the amplification resistor, as shown in Equation (3).

$$G = 4 + \frac{60 \text{ k}\Omega}{R_G}, \tag{3}$$

Once the amplifier converts the force signal into a voltage value, the data must be captured by a DAQ (NI USB 6351) and transferred to the computer. The full scale of the DAQ output has a maximum range of ±10 V. According to the 10 V input calculation, the full-scale output of the load cell is approximately 23.25 mV. Considering the full-scale value

of the DAQ, the ideal amplification factor can be inferred to be 430 times. According to Equation (3), a resistor of 140 ohms should be installed. The amplification circuit is shown in Figure 9b.

To confirm the linearity of the load cell, a weight-bearing PLA cylinder was installed above the load cell (Figure 9c). A series of weights, each weighed beforehand, were sequentially placed on top of the test structure for calibration. The average voltage recorded by the DAQ after stabilization was noted. The amplifier located at the rear end of the load cell is powered by a dual-rail 17 V input. The schematic diagram of the entire load cell measurement system is described in Figure 9d. This circuit will also be used in other measurements related to the load cell.

A total of 17 static measurements were conducted. Except for the last weight difference of 50 g, the remaining were 100 g each. The obtained raw data incorporated circuit noise. The relationship between the weight and the load cell's output was obtained by utilizing a third-order low-pass filter and averaging.

The measurement results for the 17 weights were obtained (Figure 10). The largest error among the data points compared to the regression line occurred at 1850 g, which is closest to the full scale of the DAQ. This error was 0.0275 V. By comparing it with the full scale, we found the error of linearity to be 0.275%.

| Mass(g) | Mean voltage(V) | 95% Noise(V) | Ideal voltage | Diff |
|---|---|---|---|---|
| 300 | 1.881 | 0.0052 | 1.872734 | 0.008266 |
| 400 | 2.403 | 0.0054 | 2.395234 | 0.007766 |
| 500 | 2.919 | 0.0055 | 2.917734 | 0.001266 |
| 600 | 3.43 | 0.0075 | 3.440234 | 0.010234 |
| 700 | 3.963 | 0.0066 | 3.962734 | 0.000266 |
| 800 | 4.474 | 0.0071 | 4.485234 | 0.011234 |
| 900 | 5.016 | 0.0066 | 5.007734 | 0.008266 |
| 1000 | 5.535 | 0.0069 | 5.530234 | 0.004766 |
| 1100 | 6.03 | 0.0063 | 6.052734 | 0.022734 |
| 1200 | 6.574 | 0.0077 | 6.575234 | 0.001234 |
| 1300 | 7.094 | 0.0082 | 7.097734 | 0.003734 |
| 1400 | 7.61 | 0.0079 | 7.620234 | 0.010234 |
| 1500 | 8.145 | 0.0083 | 8.142734 | 0.002266 |
| 1600 | 8.668 | 0.0085 | 8.665234 | 0.002766 |
| 1700 | 9.177 | 0.0073 | 9.187734 | 0.010734 |
| 1800 | 9.71 | 0.0081 | 9.710234 | 0.000234 |
| 1850 | 9.999 | 0.0071 | 9.971484 | 0.027516 |

(**a**)

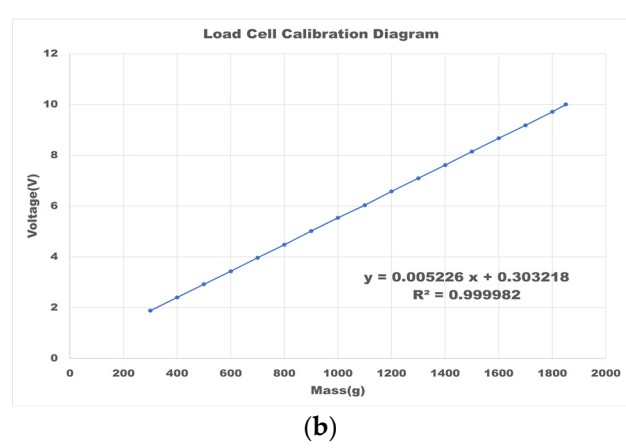

(**b**)

**Figure 10.** (**a**) Static load measurement results; (**b**) regression line plot.

### 4.3. Grinding Force Measurement

The home-made workpiece with an installed load cell to measure the grinding force was then used to perform the grinding process. Figure 11 shows the surface of the workpiece before and after the grinding process.

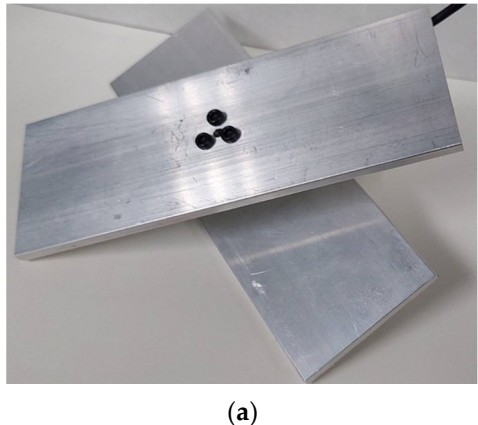

(**a**)

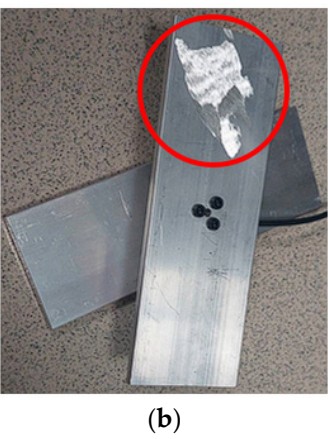

(**b**)

**Figure 11.** (**a**) Before metal grinding; (**b**) after metal grinding. The processing area is encircled by the red circle.

The force response of the load cell was recorded throughout the entire experiment, including the standby phase, four grinding phases, and three intervals between the grinding processes. Figure 12 demonstrates the normal force signal during the grinding process, measured by the load cell (setting is the same as Figure 9d) placed between two metal plates with a sampling rate of 11,000 Hz. Prior to the machining process, the workpiece is placed on the ground. The weight of the two metal plates at both ends of the load cell results in a compressive force on the load cell, making the measurement value positive. When the grinding process begins and the workpiece is lifted, the direction of the force on the load cell reverses.

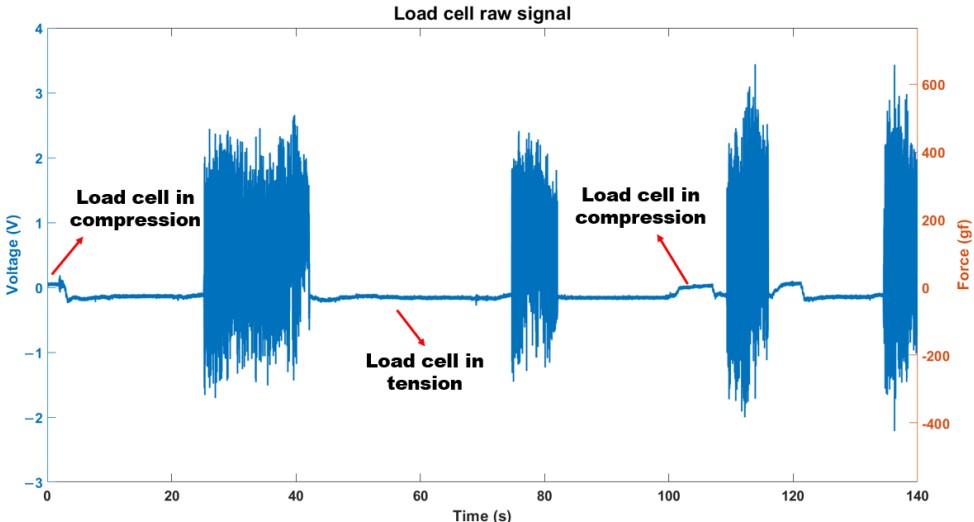

**Figure 12.** Time-domain plot of the load cell during processing.

## 5. Remote Grinding Force Sensation with a Developed Wearable Haptic System

*5.1. Frequency Domain Analysis of the Acquired Grinding Force Signal*

After we acquired the time-domain signal of the force response from the grinding process (Figure 12), further frequency-domain analysis was conducted by fast Fourier transform (Figure 13a). During machining, the MD-100's rotation speed will decrease due to friction because there is no rotational speed feedback (point 1). Additionally, we can confirm that the frequency range of the vibration can be mainly divided into low (shown at point 2), medium (point 3), and high (point 4), with the low and medium vibrations being the main components of this system.

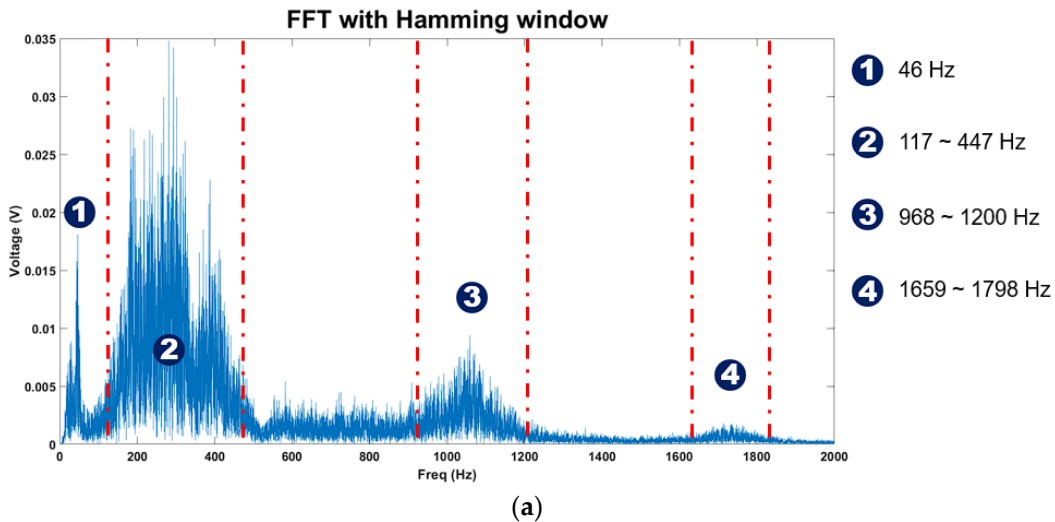

(a)

**Figure 13.** *Cont*.

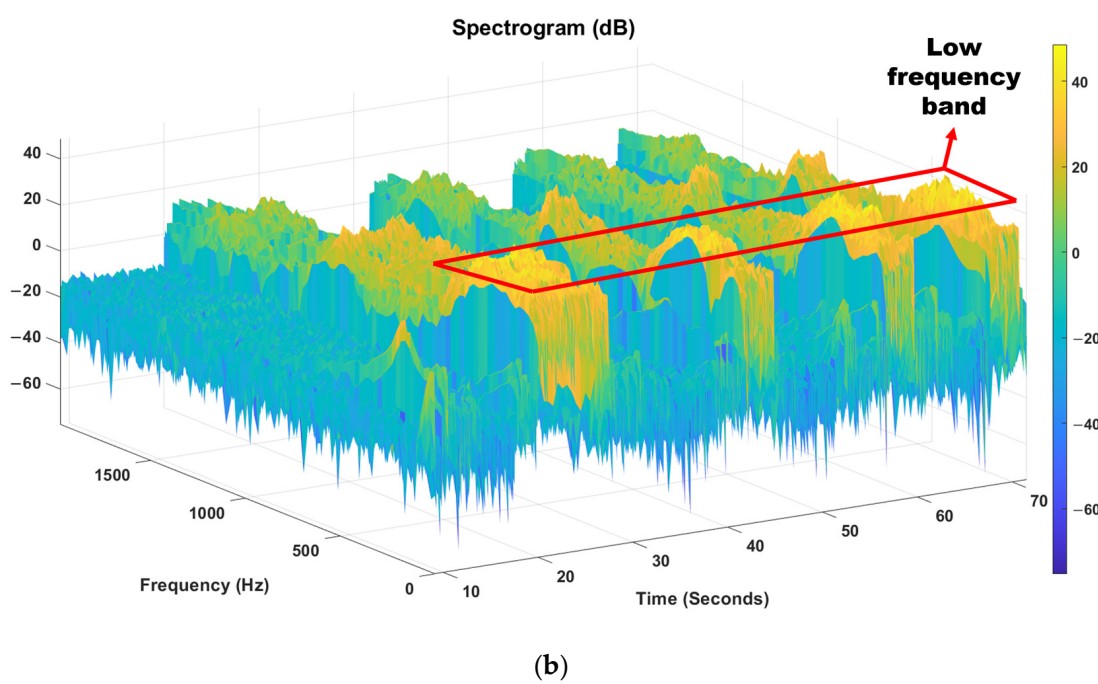

**(b)**

**Figure 13.** (**a**) Results of performing FFT from 20 to 27 s of Figure 12; (**b**) STFT diagram of Figure 12.

When the original data is processed using the Short-Time Fourier Transform (STFT) with a window length of 20 ms and an overlap of half the window length (yielding a temporal resolution of 10 ms), the time-frequency distribution of the vibrations can be obtained (Figure 13). Based on previous sections, it can be divided into low, middle, and high frequency bands. The most noticeable frequencies perceived by the user are located near the low-frequency band.

### 5.2. Methods for Transmitting Grinding Force to the Raspberry

The grinding force can be transmitted to the Raspberry in two forms (Figure 14). The first method involves transferring the recorded data file to a specific folder within the Raspberry via Wi-Fi or USB. Users can load the designated grinding force file through the CLI interface and choose whether to perform cyclic playback. The second method is applicable when the grinding machine's force measurement instrument has local network connectivity. It can connect to the Raspberry via TCP/IP. Once the connection is established, real-time data can be transmitted through this channel to the parser for the next stage of conversion. It is worth noting that if a real-time algorithm is employed, the upper and lower bounds of the force must be defined initially. This necessity arises from the requirement to delineate the maximum force, thereby enabling linear transformation. The boundaries can be extracted from the file through the utilization of the max and min functions, rendering the need for predefined limits unnecessary. In the experiment, the first method was chosen.

### 5.3. Algorithms to Convert Force Signals to PWM Command

Upon receiving force information, the RCWS server must utilize a specific algorithm to translate the force into the PWM command that will be sent to the RCWS driver. The algorithm design must take into consideration factors such as the LRA's stabilization time, data processing latency, and the effectiveness of representing vibration characteristics. The conversion process is divided into two parts: the first involves sampling the force to be converted, and the second involves transforming the sampled force signal into a PWM duty cycle through linear mapping.

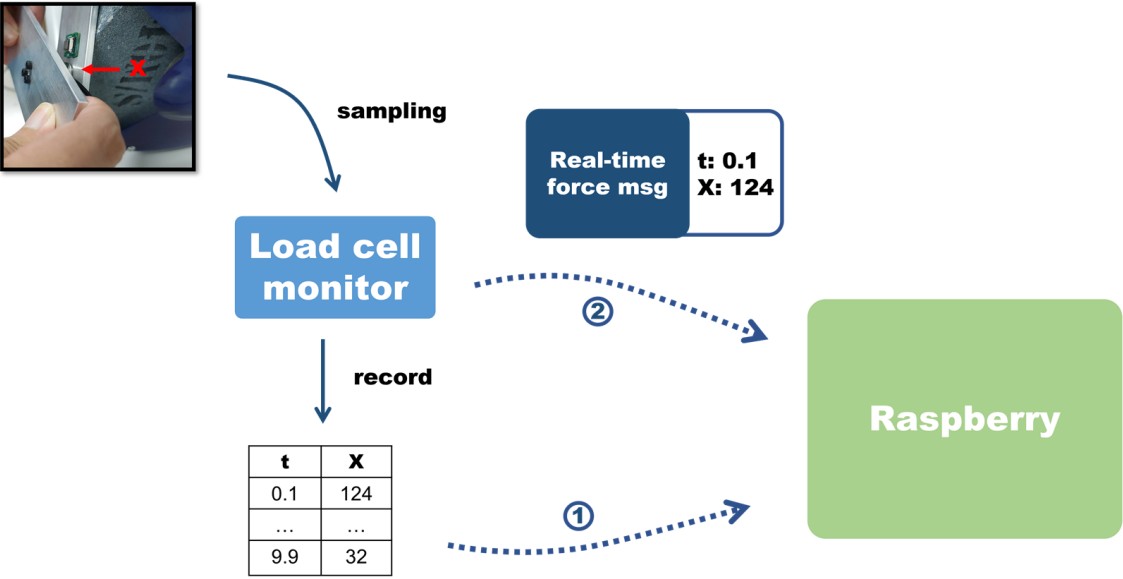

**Figure 14.** Transmitting methods for grinding machines to Raspberry.

The frequency response of the acquired grinding force was greater than 1 kHz. However, the LRA cannot adapt to such a high and wide frequency bandwidth. Therefore, it is necessary to first sample the original data in the initial stage, selecting the force signal to be processed. A fixed time interval, referred to as ΔT, is used in the following three sampling methods for the first stage:

1. Time Sampling Method (TSM): This method samples the force information acquired from the grinding workpiece at specific time points. We took the absolute value of the force signal. The resulting value can be described by the following formula:

$$F_{TSM}[k] = \left| f\left[ floor\left( \frac{t_k}{T_m} \right) \right] \right| = \left| f\left[ floor\left( \frac{k\Delta T}{T_m} \right) \right] \right| \tag{4}$$

$floor(x)$: Rounds $x$ down to the nearest integer.
$f[n]$: The discrete force signal transmitted from the machine, where $n$ is the n-th data point.
$T_m$: The sampling period of the machine.
$F_{TSM}[k]$: The force information to be passed into the quantization process.
$t_k$: The time at the k-th sampling point, $t_k = k\Delta T$.
The advantage of this method is the low latency caused by sampling, and its simplicity facilitates implementation. It is suitable for real-time processing scenarios where vibrations need to be presented to the operator. However, the sampling result is strongly correlated with ΔT, and a low sampling rate can easily lead to sampling distortion.

2. The Average Peaks Method (APM): Considering that the force during the grinding process may undergo substantial changes in a short period of time, relying solely on the TSM might neglect the significant vibration magnitudes in the selected time interval. Therefore, the APM aims to make the sampling result more consistent with the locally largest vibration values. APM forms a sequence (with $N$ values) of machine force data within ΔT, and after taking the absolute value, it extracts all the local peaks from this sequence. By averaging all the peaks, the force value sampled by APM is obtained. The sampling process within a segment ΔT can be described by the following equation:

$$f[n] = |f[n]|, \ n = 1, \ \ldots, \ N \tag{5}$$

$$P = \{f[n] \,|\, (f[n] > f[n-1] \ and \ f[n] > f[n+1]), \ n = 1, \ \ldots, \ N-1\} \tag{6}$$

$$F_{APM} = \frac{1}{|P|} \sum_{x_k \in P} x_k \tag{7}$$

$P$: The set of local peaks containing all the local maxima within the given time interval.

$N$: The total number of samples within the sequence formed in the time interval $\Delta$T.

$f[n]$: The discrete sequence representing the force data information from the machine, sampled at intervals of $T_m$.

$n$: The discrete time index represents specific instances in time where the data is sampled.

$x_k$: An element within the set $P$ representing a specific local peak.

Compared to the TSM, the APM takes into consideration the variations in data between sampling points. This could allow the sampling result to be more representative of the actual vibration trend and reduce distortion caused by random sampling (as described in Section 5.6). The averaging nature of APM smooths out the peaks in the vibration, resulting in a brief delay during sampling (specifically, one time interval $\Delta$T).

3. STFT Method: In contrast to the previous two sampling methods, this approach utilizes the STFT to extract the spectral information of the force data with a time resolution of $\Delta$T. This method simultaneously considers both the frequency and time-domain information of the vibrations.

Initially, users must select the desired frequency band of interest. The raw force data, once transformed via STFT, is used to compute the total energy within that specific band, subsequently yielding the ratio of the energy in the chosen band to the total spectral energy. Simultaneously, the raw force data is processed through the Average Peaks Method (APM) to calculate the mean force value within each time interval of length $\Delta$T. Multiplying this mean force value by the aforementioned ratio results in the final sampled force value for the selected frequency band. The sampling process within an interval $\Delta$T can be mathematically described by the following expression:

$$STFT\{f[n]\}(m, \ \omega) = \sum_{n=-\infty}^{\infty} f[n] \cdot w[n-m] e^{-i\omega n} = \sum_{n=0}^{N} f[n] \cdot w[n-m] e^{-i\omega n} \tag{8}$$

$$E_B = \sum_{\omega \in B} |STFT\{f[n]\}(\omega)| \tag{9}$$

$$E_T = \sum_{\omega} |STFT\{f[n]\}(\omega)| \tag{10}$$

$$R = \frac{E_B}{E_T} \tag{11}$$

$$F_{APM} = APM(f[n]) \tag{12}$$

$$F_{STFT} = F_{APM} \times R \tag{13}$$

$m$: The time index in the STFT, where an increment of one unit in m corresponds to a time increase of $\Delta$T.

$\omega$: The frequency in the STFT is limited by the sampling rate of the machine, and the frequency resolution ($\Delta\omega$) is determined by the number of points in the STFT.

$f[n]$: The force signal received from the machine.

$w[n-m]$: The window function, specifically utilizing the common Hamming window in this context, has an overlap of twice the number of points corresponding to the time resolution.

$B$: The set of all frequencies within the user-selected frequency band, represented at the resolution of $\Delta\omega$, encompassing all frequencies from the lower to the upper limit of the chosen range.

$E_B$: The total energy within the selected frequency bands.

$E_T$: The total energy across all frequency bands.

$R$: The ratio of energy.

For example, if the ratio is calculated based on the frequency segmentation in Figure 13a and the STFT result in Figure 13b, the distribution of the ratio over time is shown in Figure 15.

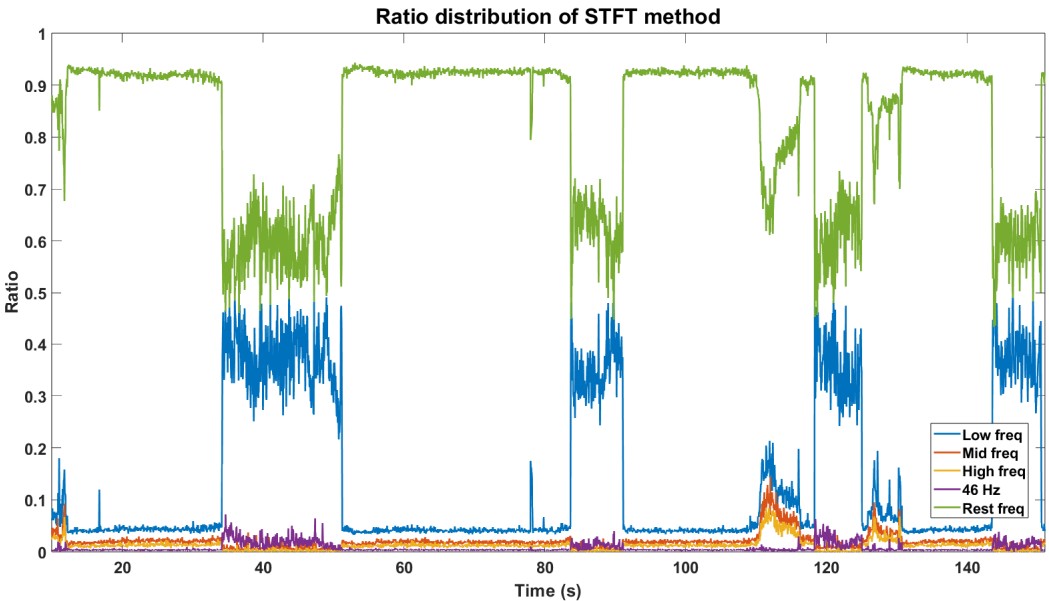

**Figure 15.** The distribution graph of the ratio over time.

In the second phase, quantization employs a linear size mapping, where the sampled force values obtained in the first phase are linearly mapped to the PWM duty cycle. This can be described as follows:

$$DC_{PWM} = a \cdot F + b, \tag{14}$$

where

$$a = \frac{DC_{PWM_{\max}} - DC_{PWM_{min}}}{F_{max} - F_{min}} \tag{15}$$

$$b = DC_{PWM_{min}} - a \cdot F_{min} \tag{16}$$

$DC_{PWM}$: PWM duty cycle, with a maximum value of 999 and a minimum value of 500 (adjustable based on actual conditions).

$F$: The sampled force, where the max and min values of the vibration must be defined by the user in real-time simulation, is automatically obtained if using a file.

### 5.4. PWM Signal Generation for the LRA Driver

In Section 5.3, PWM commands are not continuous analog signals but rather a sequence, each consisting of a specific time and duty cycle. When the designated time is reached, the Raspberry sends the command to the RCWS driver.

The LRA driver requires a continuous PWM signal, and it adjusts the AC signal output to the LRA based on the duty cycle and driving frequency.

In the STM32 MCU, the core of PWM consists of a counter, an Auto-Reload Register (ARR), and a Capture/Compare Register (CCR).

- Counter: The counter continually counts from 0 up to a specific upper limit.
- Auto-Reload Register (ARR): When the counter reaches the value in the ARR, it resets back to 0, and the state of the IO pin is toggled.

- Capture/Compare Register (CCR): When the value of the counter equals the value in the CCR, the state of the output IO pin changes.

The frequency of the PWM signal is determined by the value in the ARR, and the duty cycle is controlled by the value in the CCR. The duty cycle represents the ratio of the "on" time to the total period of the output signal.

Initially, the counter begins counting. When the counter reaches the value in the CCR, the state of the IO pin changes. When the counter reaches the value in the ARR, it resets back to 0, the state of the IO pin changes again, and the whole process starts over.

Based on the force data presented in Figure 12, a force of 385 (gf) is converted into an 80.5% duty cycle. To maintain the LRA operating at 170 Hz, the frequency of the PWM signal must be set at 21.6 kHz. To generate a PWM signal under these conditions, the following procedure is implemented:

1. The value in the Auto-Reload Register (ARR) is set to determine the entire period, corresponding to a frequency of 21.6 kHz. For the case where the clock frequency is 21.6 MHz, ARR should be set to 1000 (assuming no prescaler).
2. The value in the Capture/Compare Register (CCR) is set to control the duty cycle. In this scenario, the CCR is set to 80.5% of the ARR value, ensuring that the IO pin remains in the on state for 80.5% of the entire cycle.
3. As the counter reaches the value in the CCR, the state of the IO pin changes; when the counter reaches the value in the ARR, it resets, the process starts over, and the state of the IO pin changes again.

*5.5. Capture of Acceleration and Data Storage*

Measuring the acceleration of the vibration ring allows us to quantitatively compare the actual vibrations produced by the PWM command and the LRA and establish a corresponding relationship. Once this relationship is established, the real-time vibration acceleration feedback can be used to fine-tune the PWM values. There are several points to discuss regarding acceleration:

1. Determining the sampling rate factors
   As mentioned in Section 3, the selected accelerometer has an adjustable sampling rate. The sampling rate is mainly based on the driving frequency of the LRA. Since the frequency-acceleration peak of this LRA is around 170 Hz, if a 1000 Hz sampling rate is chosen, there would only be 6 points to describe the waveform in each cycle. This resolution is not conducive to peak acceleration analysis, so 4000 Hz was ultimately determined as the frequency for acceleration measurement.
2. Acceleration measurement range
   Through testing, it was found that the developed wearable haptic system could produce vibrations exceeding $\pm$ 8 g. Therefore, the measurement range for acceleration in this experiment was set to the maximum ($\pm$8 g).
3. Timely sampling
   Timely sampling is crucial for digital control systems. The accelerometer is equipped with a specific Data-Ready (DRDY) notification mechanism, implemented through a hardware pin interruption. This mechanism enables the Microcontroller Unit (MCU) to promptly retrieve the latest data at the instant the interruption is triggered, thus obviating the need for time-consuming polling methods. In a single-threaded bare-machine MCU environment (without the use of operating systems such as FreeRTOS), the polling approach might encounter difficulties due to potential delays caused by other processing tasks, leading to a loss of data. Conversely, the DRDY notification mechanism allows the interruption to have a higher execution priority, ensuring timely and accurate data extraction. This design not only enhances the efficiency of the system but also augments its adaptability to time-sensitive applications. Through this mechanism, the system can always obtain the latest data in the shortest possible time, ensuring accurate and consistent measurements.

4. Accelerometer data caching

   In high-frequency data collection scenarios, such as a data rate of 4000 Hz, MCUs may encounter significant data processing burdens. To effectively address this challenge, this study introduces an accelerometer data caching strategy as a core mechanism for enhancing data reading and writing efficiency. The main features of the strategy adopted in this system are as follows:

   - Data storage in a ring buffer: The circular structure of the ring buffer makes it an ideal data storage solution. Its primary advantage lies in the ability to rapidly add and remove data without the need to shift to other data. This significantly reduces data access latency, and thus, firsthand information is stored in the ring buffer after being parsed.
   - Data duplication in the main loop: To enhance data transmission efficiency, the system copies data from the Ring Buffer to the Ping Pong Buffer. This method's advantage is that by using two alternating buffers, it can reduce conflicts between read and write operations, thereby improving data transmission parallelism and throughput. This duplication process occurs during the idle periods of the main loop, avoiding blocking other operations.
   - Batch transmission strategy: To reduce transmission overhead, the system waits until a certain amount of data has accumulated, specifically 200 entries in the RCWS, and then transmits the data in the Ping Pong Buffer via USB. This strategy avoids frequent data transmission and minimizes the initialization and termination time required for each transmission.

   Through the integration of these strategies, the accelerometer data caching not only enhances data processing speed but also ensures the consistency and integrity of the data, contributing to the overall robustness and scalability of the system.

   When the user operates the RCWS device and enters DATA mode, the Raspberry side will automatically open a file to store the returned accelerometer information. The RCWS driver side will then transmit data containing the capture time and three-axis acceleration to the Raspberry according to the mechanism described in "Accelerometer data caching". Once the Raspberry has received the data, it will write the batch of data received into the file in plain text format.

*5.6. Discussion of Driven Command and Acceleration among Three Algorithms*

   The above-mentioned three different sampling methods could derive three sets of PWM commands with a ΔT of 50 ms (Figure 16). The Zero-Order Hold (ZOH) was employed to create the effective duration of the PWM commands. However, in actual implementation, the vibration only changes when a new PWM command is input, so there is no need to continuously transmit the same PWM command within ΔT.

   The Pearson correlation coefficient calculations (r) were executed to obtain the relation between the raw force signal and three sets of PWM commands. The result for $r_{TSM}$ was 0.5457, $r_{APM}$ was 0.7145, and $r_{STFT}$ was 0.7098. Among them, APM exhibited the highest degree of linear correlation, as it considered all the local peak values within ΔT. The STFT method, based on APM, resulted in slightly lower linearity. TSM had lower linearity, and this could be due to the randomness of timed sampling.

   Figure 17 shows the acceleration generated by the LRA in response to the PWM command input through the LRA driver by three different sampling methods. Upon stabilization of the LRA, we carried out a correlation coefficient calculation between the acceleration peaks and the PWM commands: $r_{TSM_{PWM-acc}}$ equal to 0.954, $r_{APM_{PWM-acc}}$ equal to 0.888, and $r_{STFT_{PWM-acc}}$ equal to 0.86. These results reveal that after the PWM commands are transmitted to the RCWS, the associated IC functions correctly. Moreover, the LRA is able to adeptly produce vibrations of corresponding amplitudes in alignment with the magnitudes of the PWM commands when stabilized. This capability leads to a high linear relationship between the PWM commands and the accelerations, demonstrating the system's precise control and responsiveness.

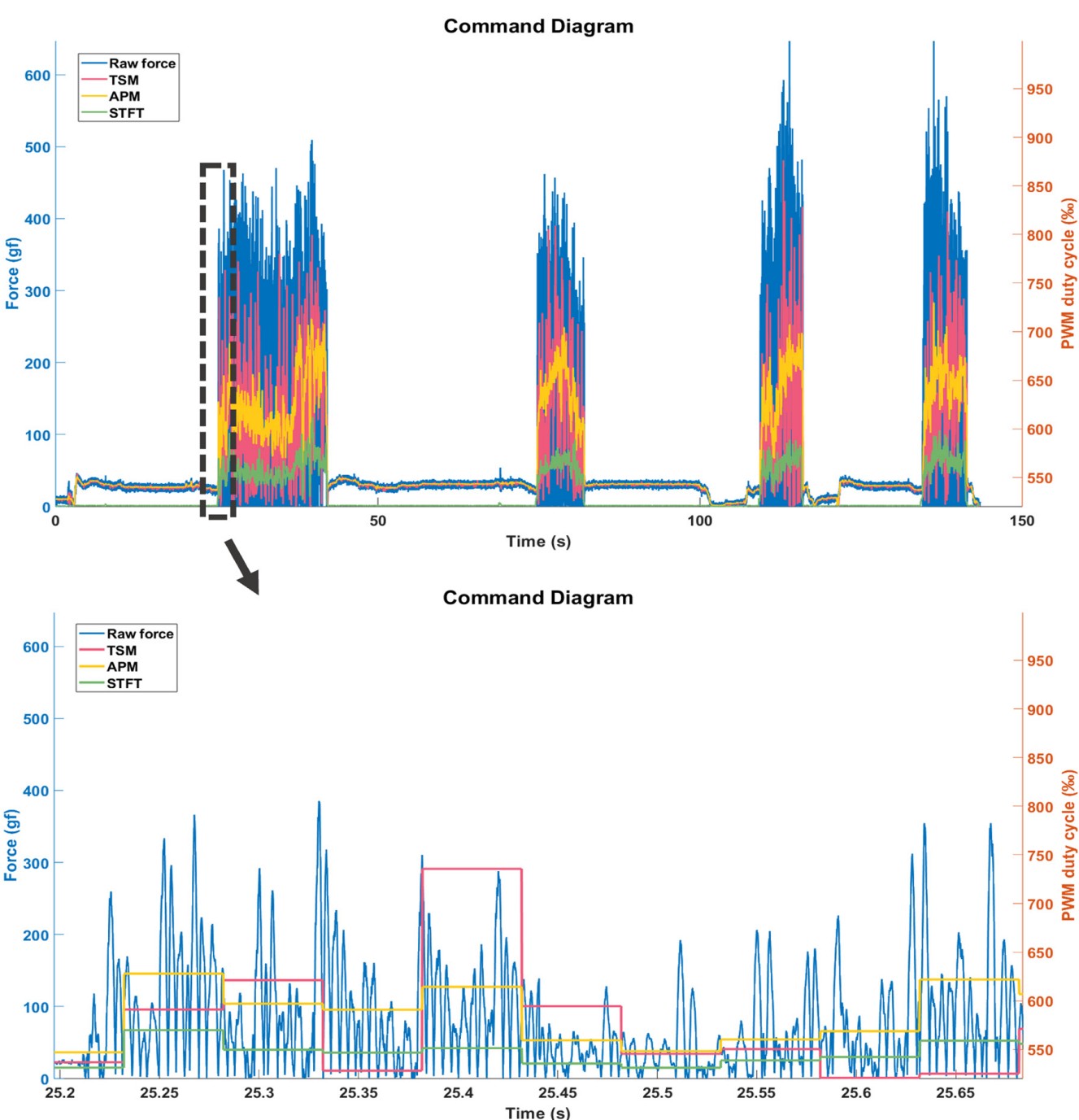

**Figure 16.** PWM command converted with ΔT = 50 ms.

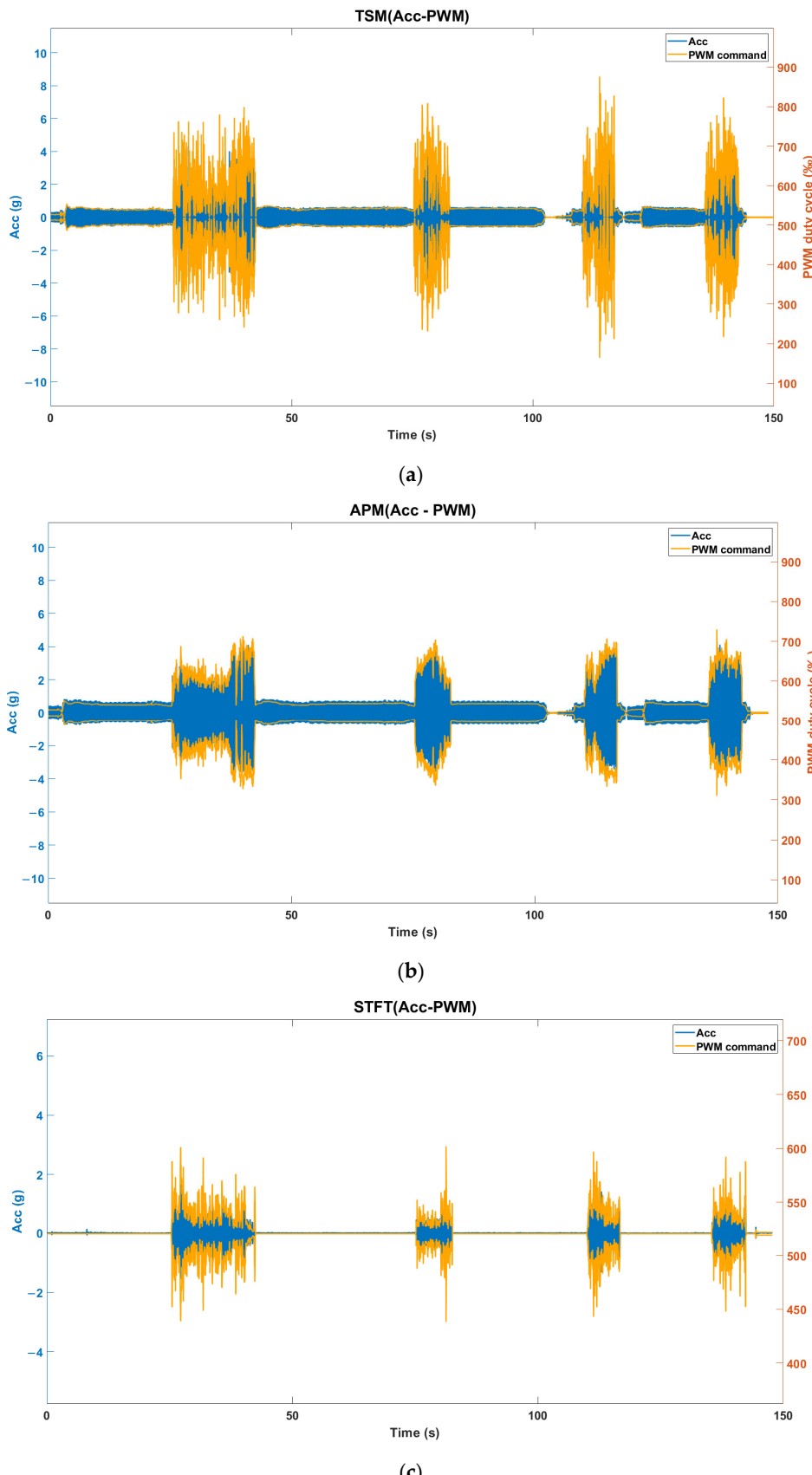

**Figure 17.** (**a**) TSM PWM-Acc diagram (ΔT = 50 ms); (**b**) APM PWM-Acc diagram (ΔT = 50 ms); (**c**) STFT PWM-Acc diagram (ΔT = 50 ms).

## 6. Conclusions

This paper presents an innovative three-axis vibration simulation system, RCWS, equipped with a three-axis Linear Resonant Actuator (LRA), specifically designed to simulate vibrations during machining processes. The built-in accelerometer in the system precisely monitors the actual vibrations generated by the LRA. In this study, we used RCWS to simulate the vibrations of manual grinding processes and utilized a loadcell system to measure the forward force endured by the workpiece during grinding. Subsequently, we employed three different methods to convert these vibration signals into PWM commands, controlling the LRA through the RCWS driver. The generated vibration effects were discussed at the end of the paper.

The paper also conducted quantitative analyses of the spectrum of manual grinding, PWM commands, and LRA characteristics. Linear correlation analysis results indicate that RCWS can simulate the magnitude of cutting forces to some extent, demonstrating potential as a human third-sense simulator in the field of smart factories and remote machining.

However, during the experimental process, we found that although the LRA has a strong vibration effect, it is constrained by the nonlinear amplitude spectrum curve, and the vibration frequency was not adjusted in this experiment. Future work could consider adopting the concept of a carrier, fixing the driving frequency of the LRA, and defining specific frequency ranges (e.g., 1~5 Hz) as the bandwidth of the modulation signal to achieve the effect of different tactile frequencies under a fixed driving frequency. Additionally, due to the longer response time of the LRA (tens of milliseconds), we could also consider using other actuators with a wider bandwidth and faster response, such as piezoelectric actuators. Paired with suitable drivers (such as the BOS1901), they can provide a higher range of vibrations and a faster response speed than the LRA. Our team plans to further explore piezoelectric tactile feedback systems in the future and integrate wireless and battery technology to upgrade RCWS into a more advanced vibration simulation device.

**Author Contributions:** Conceptualization, G.-H.F. and Y.-C.K.; methodology, S.-H.L., G.-H.F. and Y.-C.K.; software, S.-H.L.; validation, S.-H.L. and G.-H.F.; formal analysis, S.-H.L.; data curation, S.-H.L.; writing—original draft preparation, S.-H.L. and G.-H.F.; writing—review and editing, S.-H.L., G.-H.F. and Y.-C.K.; project administration, Y.-C.K. and G.-H.F.; funding acquisition, Y.-C.K. and G.-H.F. All authors have read and agreed to the published version of the manuscript.

**Funding:** This research was funded by the National Science and Technology Council in Taiwan, grant number MOST 111-2221-E-194 -053, and support from the Advanced Institute of Manufacturing with High-tech Innovations, and Department of Mechanical Engineering, National Chung Cheng University, Chiayi, Taiwan.

**Data Availability Statement:** The numerical and experimental data sets generated and analyzed during the current study are available from the corresponding author on reasonable request.

**Conflicts of Interest:** The authors declare no conflict of interest.

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
