# Peer review of "Linear Resonator Actuator-Constructed Wearable Haptic System with the Application of Converting Remote Grinding Force to Vibratory Sensation"

_actuators, doi:10.3390/act12090359_

Round 1
Reviewer 1 Report
As stated by authors (lines 75-84) in this paper they present "a tactile wearable device that integrates a three-axis commercially available LRA and MEMS accelerometer. The primary function of this device is to deliver to the grinder-beginner (or even robot) remotely (via the internet) mechanical signals accompanying manual grinding processes that have been captured during performance of the highly skilled person.
That is, indirectly authors stated that this device could be used for skill transfer (lines 11-14).
However, both the model of grinding processes and workpiece that interacts with grinding machine during manual grinding processing are able only produce occasional signals accompanying the (manual) grinding process, that is not specifically related to the highly skilled person scanpaths or specific finger-grip of holding the workpiece. Actually, haptic system is supposed to include a behavioral component what is intended to integrate tactile signals directly at accomplishing haptic exploration/rendering of grinding procedures.
The wearable device, the whole setup or even system architecture does not provide any sensor that could be able to track multiple fingertips or other manifestation of specific behavioral components of the skilled person, such as 3D position of the grinding location. So, please remove the terms which are not relevant to the paper (about skills?).
The authors are simplified the model of polishing to a single contact of workpieces with grinding wheel (e.g., Fig. 11b and Fig. 14 on the top left), by recording produced vibrations in the frequency domains what could be suitable for tactile perception according general knowledge about somatosensory perception, and recording the force applied to workpieces from the grinding wheel (this parameter would be much more valuable being recorded along the path of the contact of grinding).
Those the hand-grinding machine (shown in Fig 14) is not referenced at all, it is supposed that only used was the model MD-100 (Fig. 11b), but in such a case it is not thru that load cell shown in Fig. 9c would be suitable to capture force data properly from workpiece (Fig. 11) interacting with MD-100…
This is not clear to the reader and setup in working state configuration should be clarified.
Rendering vibration signals even modulated as a function of pressure (vector force) to grinding wheel cannot benefits technology of hand-grinding or automation.
In general, modern technologies for metal processing does not require precise haptic skills anymore.
For people/readers related to haptics the paper would have interest when replication of rendering signals would be presented to human subjects and the technique would be statistically evaluated according to standard NASA-TLX testing procedure.
Because signals processing includes many integrative procedures, linear correlation analysis (>0.7) does not prove something valuable. Technically achieving value of index >0.85 would be better.
Reviewer 2 Report
This paper is well structured and the manuscript is well written. Although the entire content is somewhat long, it seems inevitable because it covers many aspects, from the background of the research to practical applications.
